# Gangliosides and the Treatment of Neurodegenerative Diseases: A Long Italian Tradition

**DOI:** 10.3390/biomedicines10020363

**Published:** 2022-02-02

**Authors:** Maria Fazzari, Giulia Lunghi, Elena Chiricozzi, Laura Mauri, Sandro Sonnino

**Affiliations:** Department of Medical Biotechnology and Translational Medicine, University of Milano, 20129 Milano, Italy; maria.fazzari@unimi.it (M.F.); giulia.lunghi@unimi.it (G.L.); elena.chiricozzi@unimi.it (E.C.); laura.mauri@unimi.it (L.M.)

**Keywords:** ganglioside, plasma–membrane interaction, GM1 ganglioside, GM1 oligosaccharide, neurodegeneration

## Abstract

Gangliosides are glycosphingolipids which are particularly abundant in the plasma membrane of mammalian neurons. The knowledge of their presence in the human brain dates back to the end of 19th century, but their structure was determined much later, in the middle of the 1950s. From this time, neurochemical studies suggested that gangliosides, and particularly GM1 ganglioside, display neurotrophic and neuroprotective properties. The involvement of GM1 in modulating neuronal processes has been studied in detail by in vitro experiments, and the results indicated its direct role in modulating the activity of neurotrophin-dependent receptor signaling, the flux of calcium through the plasma membrane, and stabilizing the correct conformation of proteins, such as α-synuclein. Following, in vivo experiments supported the use of ganglioside drugs for the therapy of peripheral neuropathies, obtaining very positive results. However, the clinical use of gangliosides for the treatment of central neurodegeneration has not been followed due to the poor penetrability of these lipids at the central level. This, together with an ambiguous association (later denied) between ganglioside administration and Guillain-Barrè syndrome, led to the suspension of ganglioside drugs. In this critical review, we report on the evolution of research on gangliosides, on the current knowledge on the role played by gangliosides in regulating the biology of neurons, on the past and present use of ganglioside-based drugs used for therapy of peripheral neuropathies or used in human trials for central neurodegenerations, and on the therapeutic potential represented by the oligosaccharide chain of GM1 ganglioside for the treatment of neurodegenerative diseases.

## 1. Introductions

Gangliosides [1,2] are sialic acid-containing glycosphingolipids. They are inserted into the plasma membrane through the lipid fraction called ceramide and are characterized by a large hydrophilic head group that protrudes into the extracellular environment and is free to interact with neighboring molecules (lipids and/or proteins).

Their synthesis begins on the Golgi lumen, from which they are transported to the plasma membranes by vesicular transport. This compartmentalization categorizes them exclusively as components of the outer layer of membrane surface.

Gangliosides are components of all mammalian cells, but are particularly abundant in neurons, where their concentration is up to ten times higher than that of non-neuronal cells.

In neurons, gangliosides exhibit a complex pattern characterized by compounds differing in the number of sialic acid residues and structure. In human neurons, up to 5 sialic residues, present as *N*-acetyl-neuraminic acid or 9-*O*-acetyl-*N*-acetyl-neuraminic acid, are linked to position 3 of the internal and/or external galactose of the neutral chain. Disialosyl and trisialosyl chains, in which the sialic acids are linked together with a 2🡢8 bond, sometimes presenting a lactone ring Neu5Ac-(2🡢8,1🡢9)-Neu5Ac, are present in polisialylated gangliosides.

The earliest knowledge of glycosphingolipids and gangliosides dates back to the end of the 19th century [3], but their precise structure remained unknown for decades. It was in the mid-twentieth century that their structure was finally clarified [4,5,6]. This attracted the interest of many scientists, and many studies on the role played by gangliosides in human brain quickly appeared, introducing the concept that gangliosides exhibit neuroprotective and neurotrophic properties [7]. Undoubtedly, the most studied ganglioside for neuronal properties has been, and still is, the GM1 ganglioside (Figure 1) [7].

## 2. The Neuronal Plasma Membrane Organization

Among mammalian cells, neuronal plasma membranes have unique properties. Glycerophospholipids are the building blocks of the plasma membrane bilayer in all cell types, but the cholesterol content and, to a greater extent, the glycosphingolipids content, is much higher in neuronal cells. The cholesterol topology in the plasma membranes is not yet clear, but several papers state that it is largely a component of the inner layer. On the cell surface, lipids are not homogeneously distributed, but are located within membrane “domains”. Different lipid domains were characterized in terms of size, composition, and dynamics. Gangliosides are considered the membrane components that, due to their chemical, physical, and dynamic properties, drive the membrane to form lipid domains [8], together with sphingomyelin, cholesterol, and dipalmitoylphosphatidylcholine, with a low protein content. The latter usually do not exceed 2–3% of the total protein content of the cell. These domains show reduced dynamics compared with the remainder of the surface and are today known as “membrane lipid rafts” worldwide. Several receptors and protein kinases involved in the most important cell signaling processes are associated with the membrane lipid rafts and their activity is often regulated through interaction with gangliosides.

To exploit the way in which gangliosides are able to interact with and regulate membrane proteins, several experimental models were developed. Among these, the following models for modulating natural ganglioside contents deserve a mention: (i) by the addition of exogenous gangliosides to cell culture media that, under defined experimental conditions, are taken up by the cells and become components of the plasma membranes, becoming almost identical to the endogenous one; (ii) by silencing or overexpressing those enzymes involved in the ganglioside biosynthesis, both in vitro and in vivo, allowing the development of cell and animal models with altered (reduced or increased) ganglioside content.

## 3. Neurotrophic and Neuroprotective Properties of GM1 Ganglioside

To detail GM1-related functions, its plasma membrane content was increased in a variety of cultured cells, thereby modifying its membrane organization and ganglioside–protein interactions. As a result of this, the GM1 in the plasma membrane modulated the differentiation process, amplifying the response to neurotrophic factors, protecting against neurotoxicity linked to excitatory amino acid, and reducing acute nerve cell damage by blocking cytotoxicity and potentiating neurotrophic factors [9].

The earliest studies demonstrating the role of GM1 were carried out with gangliosides mixtures, where the content of GM1 was about 10%, using cholinergic and adrenergic nerve fibers that recovered from various degenerative stimuli [10,11]. In subsequent years, GM1 content was selectively modified both by its exogenous administration and by using plasma membrane sialidase Neu3 to increase the quantity of GM1. It should be remembered that Neu3 acts on all polysialylated gangliosides, causing the detachment of various sialic residues. This can lead to important changes in the ganglioside pattern, which, as a final consequence, determine the alteration of important cellular events, such as blocking the release of acetylcholine from synaptosomes and suppressing memory and learning skills [12]. It was clearly verified that GM1 promotes neurite sprouting [13,14,15,16,17,18,19,20,21,22,23,24,25,26,27] by interacting, at the plasma membrane level, with neurotrophin-dependent receptors, such as the nerve grow factor (NGF)-dependent tyrosine kinase family receptors Trk [16], and the glia cell-derived neurotrophic factor (GDNF) receptor complex RET [17,18]. Importantly, it has been proven that the presence of GM1 is necessary for neurotrophin cell signaling [19]. Its increase in the plasma membrane potentiates the process of increasing Trk kinase activity, receptor auto-phosphorylation, and dimerization in different cell types.

## 4. Gangliosides as Therapeutics

Several studies have shown that gangliosides are important regulators of cell physiology; in particular, they are regulators of neuronal homeostasis, where they regulate processes such as neuritogenesis, synaptogenesis, and cell–cell interactions. The finding that ganglioside administration to animal models of peripheral sympathetic degeneration improves the regeneration and reinnervation processes of both cholinergic and adrenergic nerve fibers suggested that gangliosides play a specific role in neural regeneration [10].

From here, several preclinical and clinical studies have been developed to demonstrate ganglioside-mediated recovery in both peripheral and central nervous system diseases. The results suggested that the administration of gangliosides achieved the following: (i) improved both electromyography parameters and clinical symptoms in patients with diabetic peripheral neuropathy; (ii) showed partial efficacy in patients with uremic neuropathy; (iii) had some neuroprotective effectiveness for acute ischemic stroke and spinal cord injuries; (iv) showed promise in the prevention and treatment of degenerative diseases of the central nervous system, particularly Parkinson’s, Alzheimer’s, and Huntington’s diseases.

## 5. The Rise to Success and the Fall of the Ganglioside-Based Drug

In 1973, the Fidia company (Abano Terme, Padua, Italy) introduced, in Italy, the drug Cronassial, containing a mixture of gangliosides derived from calf brains, and prescribed it for peripheral neuropathies [28]. It was quickly distributed, under various names, in 20 other European countries, South and Central America, Asia, and Africa. Afterwards, in 1985, the drug Sygen, containing only GM1 ganglioside, was introduced in Italy and later in Argentina, Brazil, Greece, and Singapore, and was prescribed for neurodegenerative diseases and cerebral and spinal injuries [28].

However, in the early 1990s, it was claimed that, following the administration of drugs containing gangliosides, anti-ganglioside antibodies occurred in the blood, suggesting an interaction with the axonal surface at Ranvier’s node, and suggesting that gangliosides could be responsible for the onset of some cases of Guillain-Barré syndrome (GBS) [28]. The discussion involved the Italian drug committee, which, after over 16 million prescriptions (in Italy, and many more outside of Italy, with no side effects), decided to withdraw the ganglioside-based drug at the end of 1993. The same occurred later in other countries. However, to date, Brazil and China still prescribe GM1, which is still undergoing clinical trials.

More than 30 years after the withdrawal of drugs containing gangliosides in many countries, the idea that they are not associated with serum anti-ganglioside antibodies and the onset of GBS appears to be more solid [29,30,31]. In fact, the use of gangliosides for over twenty years has never produced autoimmune diseases, GBS, or other neuropathies; some examples of controlled evidence for this include the following:A total of 5 Alzheimer patients received up to 30 mg/day of GM1 ganglioside by continuous injection into the lateral ventricles of the brain for 12 months. None of these patients developed serum antibodies that recognize gangliosides in an ELISA test. No one developed GBS or any other autoimmune neurological syndrome. Instead, these patients became more active, had improved reading comprehension, and were able to perform tasks such as writing reports and short letters on a computer [32];In 2 clinical trials, approximately 100 Parkinson patients received 100 mg of GM1 per day, either intravenously or by subcutaneous injection, for up 2 years. No one developed GBS or any other autoimmune neurological syndrome. Instead, the treatment improved motor symptoms and reduced disease progression [33];A total of 60 patients with gastrointestinal tumors were injected with 100 mg of GM1 daily for 3 days following chemotherapy. Treatment showed less protection against the oxaliplatin neurotoxicity, but no cases of GBS onset have yet been reported [34];Several studies have been conducted on over 700 patients to determine GM1 properties in the recovery of spinal cord injury, but no reports are available on the occurrence of peripheral neuropathies [35].

On the other hand, gangliosides might become immunogenic if delivered by an adjuvant, and this is the way to produce antisera in experimental animal models in most cases, with no development of any neurodegenerative disorder. There is only one article that supports the onset of an autoimmune neurologic disorder after injection of liposomes composed of gangliosides, egg lecithin, cholesterol, and methylated bovine serum albumin, emulsified in complete Freund’s adjuvant [36]. In contrast, many control sera from healthy humans, that did not show neurological symptoms, were found to contain anti-ganglioside antibodies [37]. However, anti-ganglioside antibodies injected into healthy animals did not cause the development any neurodegenerative disorders. Finally, the possible role of ganglioside contaminants in the development of neurodegenerative diseases should also be mentioned. The preparation of pure gangliosides, extracted from animal brain, requires many procedural steps to remove traces of proteins, particularly the basic myelin proteins, which are known to be encephalitogenic [38].

## 6. Blood–Brain Barrier (BBB): A Problem for Ganglioside Therapy

The therapeutic ability of gangliosides to recover damaged peripheral nervous system functions has been widely reported over the years [39,40,41]. However, clinical results for the treatment of central nervous system diseases, performed mostly with pure GM1, have been less promising due to the limited amount of ganglioside able to cross the BBB and reach target neurons when administered peripherally.

GM1, like all gangliosides, is an amphiphilic compound and forms small micelles in aqueous solutions. Below a certain concentration, called the critical micellar concentration (cmc), GM1 is present as a monomer; meanwhile, above this concentration, it is present both in the monomeric and micellar form, which are in equilibrium with each other. The cmc for GM1 is 10^−9^ M: this means that when GM1 is present in concentrations of the order of nanomolarity, or less (the concentration reached in blood in clinical trials), only nanomoles of monomers are present in solution [42]. Only GM1 in the monomeric form is able to pass, in part, through the BBB and this means that the GM1 passing through the BBB is very limited [43].

To try to overcome this problem, a high dose was used in clinical trials on Parkinson’s disease, where GM1 was administered peripherally, but without obtaining better results [33]; meanwhile, in the treatment of Alzheimer’s patients, GM1 was administered by intracranial injection, finally obtaining a beneficial effect [32]. However, this latter administration route is not compatible with an adequate quality of life.

It was also thought to chemically modify GM1, making it more permeable to the BBB passage, by replacing the acyl chain of the ceramide with a dichloroacetyl group (Figure 1) [17,44,45,46,47,48,49]. However, this compound, called LIGA20, has unfortunately been found to be toxic in the long term.

As several recent reviews report, many efforts are currently underway to find a solution for the central passage of GM1 and seem to be focused on the possibility of using liposomal vehicles [50,51,52] or intranasal infusion [53] to drive GM1 into the brain. The intranasal delivery system has been reported to reduce intracellular α-synuclein levels—enhancing expression of tyrosine hydroxylase in the substantia nigra pars compacta of the A53T α-synuclein-overexpressing mouse [53].

## 7. GM1 Oligosaccharide (OligoGM1): The Future Treatment for Neurodegenerative Diseases

Numerous clinical trials have been performed to verify GM1-mediated recovery in Parkinson’s disease (subcutaneous injection) [33], Alzheimer’s disease (intracranial administration) [32], spinal cord injury (intravenous injection) [35,36,37,38,39,40,41,42,43,44,45,46,47,48,49,50,51,52,53,54], in brain damage of premature infants (intravenous injection) [55], and in ischemic strokes of adult patients (intravenous injection) [56]. Results are difficult to correctly interpret because the expected end points have often not been reached, both due to the limited number of patients and due to the disease being at different stages among the different experimental groups.

In spite of all this, in all the human studies, positive trends have been obtained, which, in many cases, overlapped with the results obtained in the corresponding animal models.

From all these clinical studies emerges the problem of the limited accessibility of the central nervous system to GM1 [43]: in fact, this has been confirmed by the experiments carried out with the intracranial administration of GM1 both in Alzheimer’s patients [32] and in Huntington’s disease mice [57]. In both cases, the results were positive, confirming that when available in situ, GM1 is able to exert its neurotrophic and neurorestorative properties.

To solve this age-old problem, we decided to take a step back and return to the molecular mechanisms underlying the neurotrophic and neuroprotective properties of GM1, by asking a very simple question: which is the portion of the GM1 structure which is involved in neuronal processes?

Parkinson’s disease provides a good reference of a neurodegenerative disease model to understand all this, and to introduce a future therapeutic opportunity, capable of avoiding the neurodegeneration associated with GM1 ganglioside. The large majority of Parkinson’s disease (90–95%) is sporadic, with aging as the major risk factor. GM1 level, although variable among individuals, progressively decreases as we age [58,59]. Thus, in elderly people, GM1 may reach an insufficient amount to maintain dopaminergic and forebrain neurons viability, thereby gradually leading to neurodegeneration [48,59,60]. In Parkinson’s disease patients, a reduced expression of the glycosyltransferases enzymes necessary for the Golgi ganglioside synthesis, and a reduced GM1 content in the substantia nigra, in the occipital cortex [17] and in various peripheral tissues [48], were observed [17,48].

Reduction on GM1 content seems to be associated with Parkinson’s disease by two main processes: (i) the accumulation of α-synuclein due to a reduced α-synuclein–GM1 interaction; (ii) the reduction in neurotrophic signaling via neurotrophin receptors. Concerning the first point, it is necessary to remember that GM1 specifically binds tetrameric α-synuclein, promoting the α-helix protein conformation against the β-sheet-rich state of α-synuclein, whose aggregation and accumulation occurs in Parkinson’s disease [61,62,63].

Concerning the association of GM1 with the neuroprotective signaling via neurotrophin receptors, that of the TrkA has been studied in detail [20,23,26,64,65]. It is interesting to note that the absence of GM1—as it occurs in cells where the GM1 synthesis has been repressed—erases the effect exerted by NGF on the TrkA receptor [19]. This supports the information that the TrkA-GM1 complex is a prerequisite for NGF functions.

Observing in detail the crystallized structure of the TrkA-NGF complex, it is easy to note the presence of a water-occupied pocket among the two proteins. By in silico molecular docking, we demonstrated that the oligosaccharide chain of GM1 (Figure 1) perfectly fits within this space, and that the TrkA-NGF complex interaction is stabilized by OligoGM1 presence, reducing the free energy from approx. −7 kcal/mol to approx. −12 kcal/mol [66]. The engagement of the OligoGM1 in the TrkA-GM1 interaction has been confirmed by photolabeling experiments, performed with three different tritium-labeled, photoactivable GM1 derivatives (Figure 2): one photoactivable GM1 containing the reactive group at the end of the ceramide moiety, one photoactivable GM1 containing the photoactivable group at C6 of external galactose, and one photoactivable OligoGM1 containing the photoactivable group on the first galactose. Both the GM1 carrying the photoactivable group on the sugar head and the photoactivable OligoGM1 yielded a covalent complex with TrkA, while the GM1 carrying the photoactivable group at the end of ceramide did not. This suggests that the interaction occurs at the water–lipid interface, directly requiring the oligosaccharide chain of GM1, and suggests that ceramide is far from the intramembrane portion of TrkA [16]. The colocalization of GM1 and TrkA is under investigation, and it is not clear whether both belong to the same lipid domain [16,19,67]. Experiments on neuroblastoma cells, loaded with exogenous GM1, suggested that GM1 and TrkA belong to different plasma membrane regions, and that the non-raft TrkA can approach the raft GM1, allowing its outer portion to “flop” down on the membrane, bringing it into direct contact with the OligoGM1 [16].

Based on the above information, and taking into account that the neurite formation after addition of the soluble OligoGM1 to the cell medium of neuroblastoma cells was reported on many years ago [68], we began detailed studies on the role of OligoGM1 in the GM1-mediated neuronal functions.

In a short time, has been demonstrated that the neurotrophic and neuroprotective properties of OligoGM1 have an overlap with those of the entire GM1 [66,69]. Unlike GM1, its oligosaccharide, added to the cell culture medium, is not taken up by the cells, but interacts with them at the cell surface level, both in neuroblastoma and primary neurons [70].

Using murine neuroblastoma Neuro2a (N2a) cells, we demonstrated that OligoGM1 interacts with and stabilizes the TrkA-NGF complex, leading to the activation of the ERK1/2 down-streaming pathway. This event triggers cell differentiation and neuronal protection, similar to that induced by the whole GM1. By silencing or inhibiting TrkA, the differentiation and protection exerted by OligoGM1 are cancelled, suggesting that the OligoGM1–TrkA interaction is the trigger for GM1-mediated functions.

A proteomic analysis showed that OligoGM1-treated N2a cells expressed over 300 proteins, mainly associated with biochemical mechanisms of neurodifferentiation, neuroprotection, and neurorestoration (i.e., cell migration, clusterization and adhesion, activation of neuroprotective calcium signal, downregulation of pro-inflammatory molecules and glia activation, MAPK/AKT signaling). Accordingly, in cultured rat granule cells, OligoGM1 induces TrkA-MAPK activation as an early event—enhancing neuron clustering, arborization, and networking associated with the higher phosphorylation rates of FAK and Src proteins, the intracellular key regulators of neuronal motility. Furthermore, cells express increased level of neuronal markers, such as β3-Tubulin, Tau, Neuroglycan C, and Synapsin, thus anticipating the expression of complex and polysialylated gangliosides and reducing the level of simple GM3 and GD3, suggesting an advanced stage of differentiation compared with controls. We identify OligoGM1 as a mitochondrial regulator that, by acting at the plasma membrane level, triggers a biochemical signaling pathway inducing mitochondriogenesis and increasing mitochondrial activity [71]. Additionally, OligoGM1 was found to be responsible for the regulation of calcium signaling and homeostasis at the base of the neuronal functions mediated by plasma membrane GM1 [72].

In order to take advantage of the soluble OligoGM1 overcoming GM1’s pharmacological limitations (i.e., BBB penetration), we characterized the brain transport of OligoGM1 using a human in vitro BBB model [73]. OligoGM1 showed a 20-fold higher crossing rate than GM1, and time–concentration-dependent transport. Additionally, OligoGM1 crossed the barrier at 4 °C and in inverse transport experiments, allowing a consideration of the passive paracellular route. This was confirmed by the exclusion of a direct interaction with the active ATP-binding cassette (ABC) transporters. Finally, after barrier crossing, OligoGM1 remained intact and was able to induce Neuro2a cell neuritogenesis by activating the TrkA pathway. To understand whether OligoGM1 could reach the CNS, we intraperitoneally administered OligoGM1 to wild-type (WT) mice and it was found to associate with the brain, including the substantia nigra, retaining its structure [74].

Following these very promising results, the possible effects of OligoGM1 were tested on an animal model of Parkinson’s disease, based on a reduced brain content of GM1. The animal model was obtained by deleting the B4galnt1 gene, coding for the enzyme *N*-acetyl-galactosamine transferase, necessary to add a residue of *N*-acetyl-galactosamine to the ganglioside GM3, precursor of GM2 ganglioside [49]. GM2 is, in turn, the precursor of GM1 and then of the polysialylated gangliosides. In the absence of GM2, complex gangliosides are not synthesized and all GM3 is shifted to the synthesis of GD3 gangliosides series (see Figure 3).

The homozygous deletion of the B4galnt1 gene leads to the complete absence of ganglio-series, with a severe neurodegenerative phenotype of parkinsonism. On the other hand, heterozygous mice with a partial deletion of the B4galnt1 gene (B4galnt1^+/−^ mice), are characterized by a partial reduction in GM1 and GD1a gangliosides compared with wild-type mice and present a mild manifestation of Parkinson’s disease, recapitulating the progressive form of the sporadic human disease [48,49,74,75,76,77]: increase and aggregation of α-synuclein within central and peripheral neuropathological lesions, nigrostriatal degeneration, and worsening of motor and non-motor dysfunctions.

Heterozygous mice underwent replacement therapy using both GM1 and its permeable analogue LIGA20 [48,49]. This latter resulted as an effective (albeit toxic in the long term) alternative to GM1 in treatment of parkinsonian B4galnt1 GM1-deficient mice, inducing beneficial effects [48,49]. This confirmed our data that the ceramide fraction of GM1 is not critically related to the therapeutic properties exerted by GM1 and, based on this, we expected positive effects by injecting the OligoGM1 into the mouse model of Parkinson’s disease, due to its increased passage through the blood–brain barrier.

Thus, OligoGM1 was injected into adult B4galnt1^−/+^ mice [73] (20 mg/kg, intraperitoneally, daily for 28 days), completely rescuing physical symptoms, reducing α-synuclein aggregates, and restoring tyrosine hydroxylase expression and neurotransmitter levels in substantia nigra, thus matching the wild-type healthy condition.

## 8. Conclusions

Research on brain lipids began in the late 19th century, providing the first, very rough, information on the existence of gangliosides [3]. It took another half a century to understand what gangliosides were from the chemical point of view. From here, their specific role in modulating and protecting brain functions was quickly understood (Figure 4).

This has sparked great interest in the use gangliosides as drugs to prevent and reduce neurodegenerative mechanisms. Positive therapeutic results were obtained by treating peripheral neuropathies with ganglioside-based drugs, while the results regarding degenerative diseases of the central nervous system were less exciting. This is due to the chemical–physical properties of gangliosides: they are amphiphilic molecules that aggregate in aqueous solutions, forming high molecular weight micelles that are unable to pass the BBB.

Finally, it has been experimentally clear that the neuroprotective and neurotrophic properties exerted by GM1 ganglioside, widely demonstrated in in vitro and in vivo experiments, are associated with its oligosaccharide portion, which could be considered a valid structure for the preparation of drugs. The removal of the ceramide tail from GM1, which allows to the oligosaccharide to cross efficiently through the BBB and to gain good solubility and stability, and the possibility for it to be synthetized without animal use (i.e., bacterial synthesis)—eliminating any possible protein contamination—are all advantages in the development of a potential oligosaccharide-based drug [78,79,80]. Finally, taking into account the results obtained by injecting GM1 directly into human Alzheimer’s patients’ brains or Huntington’s mice brains, we believe that the GM1 oligosaccharide could soon become an effective drug for neurodegenerative and neurodevelopmental diseases where neurotrophic membrane signaling appears to be impaired. Studies on animal models of Alzheimer’s disease, Huntington’s disease, LAS, and Rett syndrome are underway. In parallel, we are developing the OligoGM1 preparation on an industrial scale by a bacterial route, with the hope that it will soon be a real therapeutic drug. 

## Figures and Tables

**Figure 1 biomedicines-10-00363-f001:**
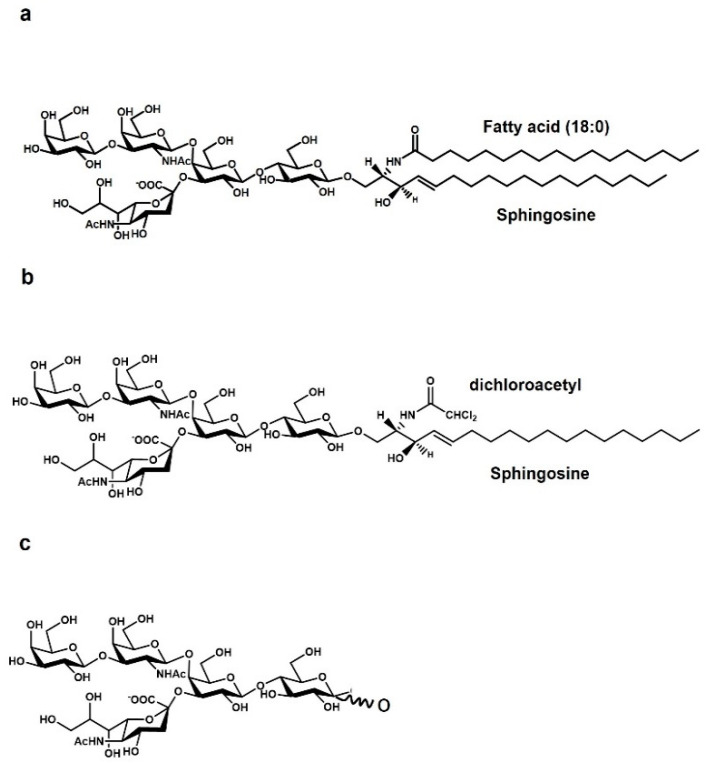
Chemical structure of (**a**) GM1 ganglioside, (**b**) the GM1 permeable analogue LIGA 20, and (**c**) the GMI oligosaccharide chain, the OligoGM1.

**Figure 2 biomedicines-10-00363-f002:**
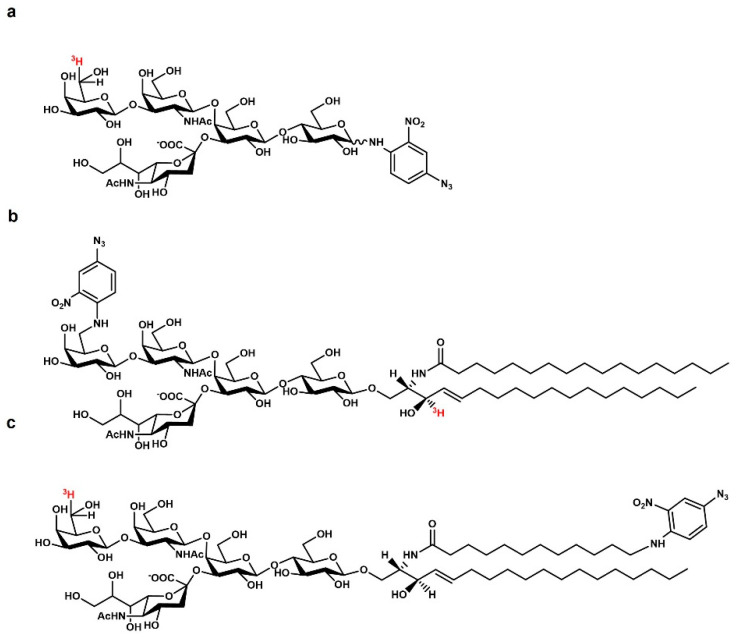
Chemical structure of (**a**) the tritium-labeled, photoactivable compounds. OligoGM1 containing tritium at C6 of the external galactose and the nitrophenyl azide at C1 of glucose; (**b**) GM1 ganglioside containing the nitrophenyl azide at C6 of galactose and tritium at C3 of sphingosine; (**c**) GM1 ganglioside containing tritium at C6 of external galactose and nitrophenyl azide at the end of the acyl chain.

**Figure 3 biomedicines-10-00363-f003:**
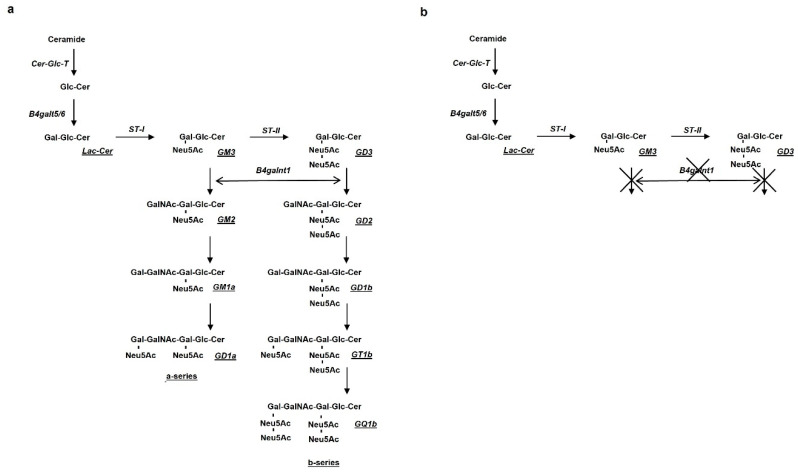
Steps of the biosynthesis of the main brain gangliosides: (**a**) in the absence of B4galnt1, the biosynthesis of GM2 cannot proceed and (**b**) the process stops with GM3 and GD3.

**Figure 4 biomedicines-10-00363-f004:**
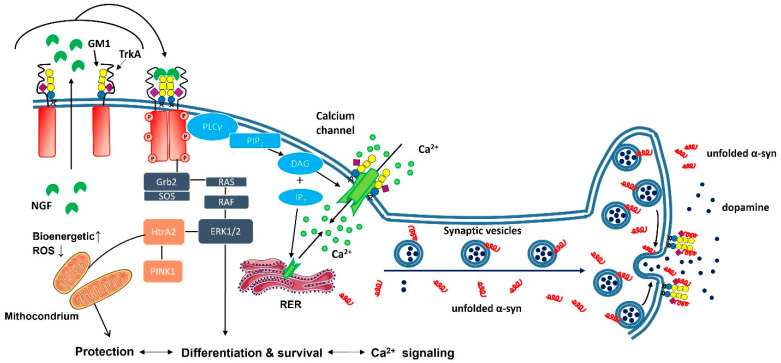
Neuronal processes modulated by plasma membrane GM1. GM1 is a component of the outer layer of the plasma membrane and when available in physiological amounts it interacts with the NGF receptor TrkA. This interaction is necessary for TrkA dimerization and activation when NGF is released by the cell. From here, the process activates the down-streaming pathway, finally leading to the differentiation, protection, and survival of neurons. Gangliosides around the calcium channel concentrate calcium ions to realize a strong gradient and accelerate calcium’s entrance into the cell. The GM1 associated with the synapsis stabilizes the α-synuclein, avoiding its aggregation, and allowing it to transfer to the post-synapsis as an unfolded protein.

## Data Availability

Not applicable.

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
