# Peer review of "Gangliosides and the Treatment of Neurodegenerative Diseases: A Long Italian Tradition"

_biomedicines, 2022, doi:10.3390/biomedicines10020363_

Round 1
Reviewer 1 Report
The review submitted by Fazzari and coll., regarding the gangliosides, is well written. experiments in vivo (animal model and human brain) and in vitro confirm the activity of oligosaccharide portion of ganglioside GM1as neuroprotective and neurotrophic properties.Therefore, it is hoped that
ganglioside can be industrially prepared as a drug for the treatment of
neurodegenerative diseases. This review could be published in present form.
Author Response
We thank the Reviewer for the positive evaluation.
Reviewer 2 Report
The authors are trying to summarize evidences of ganglioside functions for neurodegenerative diseases. The aim of this review is quite interesting and very important. Overall, the coverage of this topic is broad but appears somewhat superficial. There are several concerns which should be addressed by the authors.
It would be appropriate if the authors provide original concept for this review to be distinguished from other authors’ articles. What is the originality and novelty in this review? The authors need to describe the author's remarkable idea and outstanding future direction as an original review article.
Gangliosides are difficult to penetrate through the blood-brain barrier. Recently, Revunov et al., (2020) reported that less than 0.4% of intravenously injected of a radiofluorinated ganglioside, [18F]F-GM1, could enter the non-human primate brain. Amount of chemicals exposed to the brain by intraperitoneal, subcutaneous or intramuscular injection is typically less than that of intravenous injection. Since delivery of gangliosides is extremely small amount even by intravenous administration, it is important to choose a more efficient route for ganglioside administration for therapy. Subcutaneous, intravenously or intramuscular injection of GM1 has been tried with limited success or failed. So far, intracerebroventricular administration is the most reliable method to deliver gangliosides into the brain. It would be helpful to describe the route of delivery and the limitations of each infusion methods on all reported studies, although routes of some research are mentioned. Kumbale et al (1999), Sukhov et al. (2020), and Itokazu et al. (2021), evidenced that GM1 can be delivered to the brain and the spinal cord ​via the olfactory neural pathway. Citing those references would be of great service to the readers in the context of the route of gangliosides to CNS.
The safety profile of GM1 trials were impressive with no serious adverse events reported and no clinically significant abnormalities in blood chemistry, hematology measures, or urinalysis results. There were no antigenicity by ganglioside treatment. The authors mentioned that “the administration of drugs containing gangliosides, anti-ganglioside antibodies occurred in blood, suggesting their interaction with the axonal surface at the Ranvier’s node and that gangliosides could be responsible for the onset of some cases of Guillain-Barré syndrome". How this hypothesis was completely wrong should be discussed more in this review. As the authors mentioned, ganglioside itself is not antigenic and could not produce antibodies without adjuvant treatment. Detailed discussion would be helpful to confirm the safety and tolerability of gangliosides for treatments.
The purity of gangliosides is the most important. It should be emphasized that the purity of the ganglioside preparations has not been ascertained in any of the previous reports. Even some studies did not provide source of gangliosides that used for experiments or trials. It needs to be extremely careful to judge their ganglioside purities and sources. In fact, previous studies have shown that intensive sensitization of rabbits with gangliosides does not produce any apparent neurological symptoms, despite the fact that high-titer antibodies can be elicited. The discrepancy may be due to the fact that gangliosides isolated from nerve tissues typically are contaminated by proteins and require special steps in purification to remove the last trace of proteins, especially contamination of myelin basic proteins that are notoriously encephalitogenic. Most of these studies with negative effects of gangliosides were owing largely to the use of gangliosides of questionable purity, particularly from contamination by proteins. Extreme care, therefore, must be exercised in assessing the purity of the ganglioside preparations to ensure that they are free of any proteins. Studies with unclear source or purity of ganglioside cannot be reproducible nor reliable.
The authors mentioned Neu3 to amplify GM1. Since Neu3 break more complex gangliosides, it might be worse outcomes. It has been reported that treatment of anti-GT1aα/GQ1bα antibody blocks the release of acetylcholine from synaptosomes, and the memory and learning abilities are remarkably suppressed (Ando et al., 2004). It would be desirable to discuss this point for safely utilizing sialidase.
As the authors have studied GM1 oligosaccharide for long time, it would be helpful to discuss the limitation of carbohydrate portion without sphingolipids.
Line 435, Reference 27: “Barrpè” should be “Barrè”.
Author Response
1) It would be appropriate if the authors provide original concept for this review to be distinguished from other authors’ articles. What is the originality and novelty in this review? The authors need to describe the author's remarkable idea and outstanding future direction as an original review article.
We partially modified the abstract to make more clear that this review deals with a critic analysis of the knowledge on gangliosides as molecules involved in biological processes and as therapeutic drugs.
2) Gangliosides are difficult to penetrate through the blood-brain barrier. Recently, Revunov et al., (2020) reported that less than 0.4% of intravenously injected of a radiofluorinated ganglioside, [18F]F-GM1, could enter the non-human primate brain. Amount of chemicals exposed to the brain by intraperitoneal, subcutaneous or intramuscular injection is typically less than that of intravenous injection. Since delivery of gangliosides is extremely small amount even by intravenous administration, it is important to choose a more efficient route for ganglioside administration for therapy. Subcutaneous, intravenously or intramuscular injection of GM1 has been tried with limited success or failed. So far, intracerebroventricular administration is the most reliable method to deliver gangliosides into the brain. It would be helpful to describe the route of delivery and the limitations of each infusion methods on all reported studies, although routes of some research are mentioned. Kumbale et al (1999), Sukhov et al. (2020), and Itokazu et al. (2021), evidenced that GM1 can be delivered to the brain and the spinal cord via the olfactory neural pathway. Citing those references would be of great service to the readers in the context of the route of gangliosides to CNS.
We thank the Reviewer for the suggestion and added the following recerence: “Yutaka Itokazu, Takahiro Fuchigami, John C Morgan, Robert K Yu Intranasal infusion of GD3 and GM1 gangliosides downregulates alpha-synuclein and controls tyrosine hydroxylase gene in a PD model mouse. Mol Ther 2021;29:3059-3071”
3) The safety profile of GM1 trials were impressive with no serious adverse events reported and no clinically significant abnormalities in blood chemistry, hematology measures, or urinalysis results. There were no antigenicity by ganglioside treatment. The authors mentioned that “the administration of drugs containing gangliosides, anti-ganglioside antibodies occurred in blood, suggesting their interaction with the axonal surface at the Ranvier’s node and that gangliosides could be responsible for the onset of some cases of Guillain-Barré syndrome". How this hypothesis was completely wrong should be discussed more in this review. As the authors mentioned, ganglioside itself is not antigenic and could not produce antibodies without adjuvant treatment. Detailed discussion would be helpful to confirm the safety and tolerability of gangliosides for treatments.
We added a short paragraph to explain on the ineffectiveness of the anti-ganglioside antibodies circulating in blood in promoting neurodegenerative disorders. We added the following reference “Nobile-Orazio E., Carpo M. and Legname G., Meucci,N., Sonnino,S. and Scarlato,G. (1990) Anti-GM1IgM antibodies in motor neuron disease and neuropathy. Neurology, 40, 1747-1750”
4) The purity of gangliosides is the most important. It should be emphasized that the purity of the ganglioside preparations has not been ascertained in any of the previous reports. Even some studies did not provide source of gangliosides that used for experiments or trials. It needs to be extremely careful to judge their ganglioside purities and sources. In fact, previous studies have shown that intensive sensitization of rabbits with gangliosides does not produce any apparent neurological symptoms, despite the fact that high-titer antibodies can be elicited. The discrepancy may be due to the fact that gangliosides isolated from nerve tissues typically are contaminated by proteins and require special steps in purification to remove the last trace of proteins, especially contamination of myelin basic proteins that are notoriously encephalitogenic. Most of these studies with negative effects of gangliosides were owing largely to the use of gangliosides of questionable purity, particularly from contamination by proteins. Extreme care, therefore, must be exercised in assessing the purity of the ganglioside preparations to ensure that they are free of any proteins. Studies with unclear source or purity of ganglioside cannot be reproducible nor reliable.
We completely agree with the Reviewer’s comment and added a phrase on this in the paragraph “Success and fall of the ganglioside based-drug”
5) The authors mentioned Neu3 to amplify GM1. Since Neu3 break more complex gangliosides, it might be worse outcomes. It has been reported that treatment of anti-GT1aα/GQ1bα antibody blocks the release of acetylcholine from synaptosomes, and the memory and learning abilities are remarkably suppressed (Ando et al., 2004). It would be desirable to discuss this point for safely utilizing sialidase.
In the paragraph “Neurotrophic and neuroprotective properties of GM1 ganglioside” we added a comment on the effects of desialylation by Neu3 and on the negative effects possibly exerted on synaptic functions. We also added the the reference Ando S, Tanaka Y, Kobayashi S, Fukui F, Iwamoto M, Waki H, Tai T, Hirabayashi Y. Synaptic function of cholinergic-specific Chol-1alpha ganglioside. Neurochem Res. 2004;29:857-67
6) As the authors have studied GM1 oligosaccharide for long time, it would be helpful to discuss the limitation of carbohydrate portion without sphingolipids.
We do not see any limitation in the use of the GM1 oligosaccharide rather than the ganglioside GM1. The removal of the lipid moiety increasing the passage through the BBB. We added a short comment on this
7) Line 435, Reference 27: “Barrpè” should be “Barrè”.
We fixed the typo.
Round 2
Reviewer 2 Report
The reviewer appreciates that authors have given well answered in the letter and the revised manuscript. The paper has been improved very much. The manuscript is now acceptable for publication.